# Scalable Deep Neural Networks via Low-Rank Matrix Factorization

## Abstract

Compressing deep neural networks (DNNs) is important for real-world applications operating on resource-constrained devices. However, it is difficult to change the model size once the training is completed, which needs re-training to configure models suitable for different devices. In this paper, we propose a novel method that enables DNNs to flexibly change their size after training. We factorize the weight matrices of the DNNs via singular value decomposition (SVD) and change their ranks according to the target size. In contrast with existing methods, we introduce simple criteria that characterize the importance of each basis and layer, which enables to effectively compress the error and complexity of models as little as possible. In experiments on multiple image-classification tasks, our method exhibits favorable performance compared with other methods.

## 1 Introduction

As part of the great progress made in deep learning, deep neural network (DNN) models with higher performance have been proposed for various machine-learning tasks (LeCun et al., 2015). However, these performance improvements require a higher number of parameters and greater computational complexity. Therefore, it is important to compress them without sacrificing the performance for running the models on resource-constrained devices.

Han et al. (2016) reduced the memory requirement for devices by pruning and quantizing weight coefficients after training the models. Howard et al. (2017); Sandler et al. (2018); Howard et al. (2019) used factorized operations called depth-wise and point-wise convolutions in a proposal for light-weight models suited to mobile devices. However, these methods require pre-defined network structures and pruning the model weights after training. Recently, automated frameworks, such as the so-called *neural architecture search (NAS)* (Zoph & Le, 2017), have been proposed. Tan et al. (2019) proposed a NAS method to accelerate the inference speed on smartphones by incorporating resource-related constraints into the objective function. Stamoulis et al. (2019) significantly reduced the search costs for NAS by applying a gradient-based search scheme with a *superkernel* that shares weights for multiple convolutional kernels.

However, the models trained by these methods are dedicated to specific devices, and thus do not possess the ability to be reconfigured for use on different devices. In order to change the model size, it is necessary to re-train them according to the resources of the target devices. For example, it has been reported that the inference speed when operating the same model on different devices differs according to the computing performance and memory capacity of the hardware accelerator (Ignatov et al., 2018). Therefore, it is desirable that the model size can be flexibly changed according to the resources of the target devices without re-training the model, which we refer to as scalability in this paper.

To this end, Yu et al. (2019) introduced switchable batch normalization (BN) (Ioffe & Szegedy, 2015), which switches BN layers according to pre-defined widths, and proposed "slimmable" networks whose width can be changed after training. Moreover, Yu & Huang (2019) proposed universally slimmable networks (US-Nets) that extend slimmable networks to arbitrary widths. However, since these methods directly reduce the width (i.e., dimensionality) in each layer, the principal components are not taken into account. In addition, they reduce the width uniformly across all layers, which ignores differences in the importance of different layers.

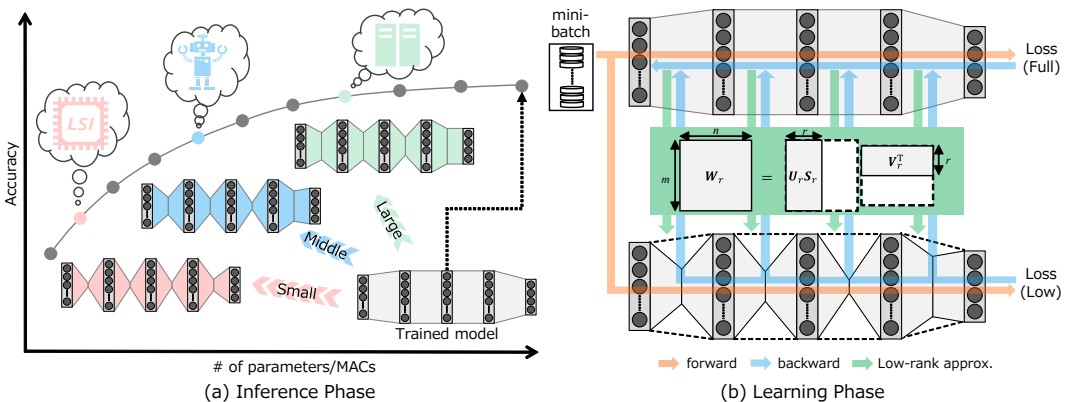

(a) Inference Phase         (b) Learning Phase

Figure 1: Illustration of our scalable neural networks. (a) Inference phase of our method. The model can scale according to the target size by changing the rank of a weight matrix in each layer. (b) Learning phase of our method. We explicitly minimizes losses for both of full- and low-rank network, which is designed not only to keep the performance of full-rank network but also to improve multiple low-rank networks (to be used at the inference phase). These schemes can also be applied to CNNs.

In this paper, we propose a novel method that enables DNNs to flexibly change their size after training. We factorize a weight matrix in each layer into two low-rank matrices after training the DNNs via singular value decomposition (SVD). By changing the rank in each layer, our method can scale the model to an arbitrary size (Figure 1(a)). Our contributions are as follows.

- We do not directly reduce the width but instead reduce the redundant basis in the column space of the weight matrix, which prevents the feature map in each layer from losing important features.

- We introduce simple criteria that characterize the importance of each basis and layer, namely, the error- and complexity-based criteria. These enable to effectively compress the error and complexity of the models as little as possible.

- We facilitate the performance of low rank networks with the following methods: a learning procedure that simultaneously minimizes losses for both the full and low rank networks (Figure 1(b)), and the mean & variance correction for each BN layer according to the given rank.

In the experiments on image-classification tasks of the CIFAR-10/100 (Krizhevsky, 2009) datasets using deep convolutional neural networks (CNNs), our method exhibits better performance for up to approximately 75% compressed models than slimmable networks and US-Nets. In the following, we first describe the details of our method (Section 2) and briefly review related works (Section 3). Then, we give some experimental results (Section 4) and conclude the paper (Section 5).

## 2 METHODS

In this section, we first give an overview then describe the details of the inference and learning methods.

### 2.1 OVERVIEW

For a layer in the network, let $\boldsymbol{y} = \boldsymbol{W}^{\top}\boldsymbol{x} \in \mathbb{R}^n$ be an output vector given by linear transformation of an input vector $\boldsymbol{x} \in \mathbb{R}^m$ with a weight matrix $\boldsymbol{W} \in \mathbb{R}^{m \times n}$, where $m$ and $n$ are the numbers of input and output nodes, respectively. Let $R$ be the rank of the weight matrix, with $1 \leq R \leq \min(m, n)$. Given $\boldsymbol{U} = (\boldsymbol{u}_1, \ldots, \boldsymbol{u}_R) \in \mathbb{R}^{m \times R}$ and $\boldsymbol{V} = (\boldsymbol{v}_1, \ldots, \boldsymbol{v}_R) \in \mathbb{R}^{n \times R}$ as matrices that have left and right singular vectors (i.e., bases) in columns, and $\boldsymbol{S} = \text{diag}(\sigma_1, \ldots, \sigma_R) \in \mathbb{R}^{R \times R}$ as a matrix composed of singular values in diagonal components, we can formulate the SVD as $\boldsymbol{W} = \boldsymbol{U}\boldsymbol{S}\boldsymbol{V}^{\top}$.

An example of our scalable neural networks with fully connected layers is shown in Figure 1. After the training, each weight matrix in the network is factorized into two matrices of rank $R$ via SVD, and we control this value to change the model size. This can be viewed as inserting a sub-layer between the original layers and changing its width $R$. For the convolutional tensor $\mathbf{W} \in \mathbb{R}^{K_w \times K_h \times C_{in} \times C_{out}}$ of kernel width $K_w$, kernel height $K_h$, input channels $C_{in}$, and output channels $C_{out}$, we transform it to the matrix form $\boldsymbol{W} \in \mathbb{R}^{K_w K_h C_{in} \times C_{out}}$, where $m = K_w K_h C_{in}$ and $n = C_{out}$. Then, we apply SVD as in Zhang et al. (2016); Wen et al. (2017). This yields two layers with a tensor $\mathbf{W}_1 \in \mathbb{R}^{K_w \times K_h \times C_{in} \times R}$ and a tensor $\mathbf{W}_2 \in \mathbb{R}^{1 \times 1 \times R \times C_{out}}$. The number of parameters in each layer becomes $(m + n) R$ by this factorization. Thus, we can compress the network to an arbitrary size by changing the rank $r$ $(\leq R)$ within the range $1 \leq r < mn/(m + n)$.

Associated with changing the rank, the monotonicity of approximation error holds for each layer.

**Proposition 2.1.** *Let $\boldsymbol{W}_r = \boldsymbol{U}_r \boldsymbol{S}_r \boldsymbol{V}_r^{\top}$ be a rank-r approximation using the truncated-SVD for $\boldsymbol{W}$ and let $\boldsymbol{y}_r = \boldsymbol{W}_r^{\top} \boldsymbol{x}$. Then, the squared error between an original $\boldsymbol{y}$ and $\boldsymbol{y}_r$ satisfies that $\|\boldsymbol{y} - \boldsymbol{y}_1\|^2 \geq \cdots \geq \|\boldsymbol{y} - \boldsymbol{y}_r\|^2 \geq \|\boldsymbol{y} - \boldsymbol{y}_{r+1}\|^2 \geq \cdots \geq \|\boldsymbol{y} - \boldsymbol{y}_R\|^2 = 0$.*

The proof is given in Appendix A. According to the above, errors between an original output $\boldsymbol{y}$ and its approximation $\boldsymbol{y}_r$ monotonically decrease as the rank increases. Hence, it can be expected the performance of the entire network will scale with the model size, which is controlled by the rank in our method.

## 2.2 INFERENCE

### 2.2.1 RANK SELECTION

Given a target size for a model, we select the rank of each layer by reference to the following criteria.
**Error-based criterion**. According to Eq. (5) in Appendix A, the error associated with a rank-1 decrease is given by $|\boldsymbol{v}^{\top} \boldsymbol{y}| = \sigma |\boldsymbol{u}^{\top} \boldsymbol{x}| = \sigma \|\boldsymbol{x}\| |\cos \theta|$. This implies that the error depends on the singular value $\sigma$ and the cosine similarity between an input vector $\boldsymbol{x}$ and the corresponding left singular vector $\boldsymbol{u}$. Based on this, we consider how to compress the model with as little error as possible by reducing the bases that induce lower errors. It has been reported that networks with BN layers and ReLUs (rectified linear units) (Glorot et al., 2011) possess the scale-invariance property (Arora et al., 2019). Thus, the error $|\boldsymbol{v}^{\top} \boldsymbol{y}|$ should be normalized by the scale of $\boldsymbol{y}$ in each layer. Exploiting the fact that $\|\boldsymbol{y}\| \leq \|\boldsymbol{W}\|_2 \|x\|$, we normalize it as $|\boldsymbol{v}^{\top} \boldsymbol{y}| / (\|\boldsymbol{W}\|_2 \|x\|) = \sigma |\cos \theta| / \|\boldsymbol{W}\|_2 \in [0, 1]$, where $\|\boldsymbol{W}\|_2$ is the spectrum norm of $\boldsymbol{W}$ (i.e., the maximum singular value).

Computing the cosine similarities is costly because each layer requires input $\boldsymbol{x}$ over the whole dataset. Therefore, we omit it and simply use the following criterion for selecting the rank:

$$C_1(\ell, k) = \sigma_k^{(\ell)} / \|\boldsymbol{W}^{(\ell)}\|_2, \tag{1}$$

where $\ell$ is a layer index. This is equivalent to keeping $\|\boldsymbol{W}^{(\ell)} - \boldsymbol{W}_r^{(\ell)}\|_F^2$ small in each layer. We consider this is a simple but effective criterion for the following reasons. First, Arora et al. (2018) have reported that the subspace spanned by each layer's weight vectors and the subspace spanned by their input vectors both become implicitly low rank and correlated after training. In other words, there should be many small singular values in each layer's weight matrix. Second, the principal directions of the weights are correlated with those of the inputs. Thus, by reducing the bases that correspond to smaller singular values, we can reduce by a large number of ranks without significantly increasing the errors. Moreover, the cosine similarities are expected to be higher for large singular values, meaning that our method can reflect the principal directions of the data distribution even if we only use the singular values of the weight matrices as the criterion.

**Complexity-based criterion**. We achieve a high compression rate by reducing the rank in layers that have a large number of parameters and multiply-accumulate operations (MACs). For convolutional layers, the numbers of parameters, excluding biases and the MACs, are given by $P = K_w K_h C_{in} C_{out}$ and $M = PHW$ for a feature map of height $H$ and width $W$, respectively.

We use the following as a complexity-based criterion:

$$C_2(\ell) = \left(1 - P^{(\ell)} / \sum_i P^{(i)}\right)\left(1 - M^{(\ell)} / \sum_i M^{(i)}\right), \tag{2}$$

where $P^{(\ell)}$ and $M^{(\ell)}$ are the numbers of parameters and the MACs in layer $\ell$, respectively. By coupling the above two criteria, we reduce the bases with lower values of $C(\ell, k) = C_1(\ell, k)\, C_2(\ell)$ across the entire network. In practice, we compute the criterion for all bases after training. Then, we sort them in ascending order and store as a list. The only necessary step for selection is to reduce the first $d$ bases in the list, where $d$ is determined by the target model size. The algorithm is given in Appendix B.

### 2.2.2 BN CORRECTION

As pointed out by Yu et al. (2019), the means and variances of the BN layers should be corrected when the model size is changed. Suppose that a BN layer is inserted right after the convolutional layer, and that the mean and variance of $\boldsymbol{y}\left(= \boldsymbol{W}^\top \boldsymbol{x}\right)$ are normalized in the BN layer. Then, we should correct those values according to the rank-$r$ approximation of $\boldsymbol{y}$ (i.e., $\boldsymbol{y}_r$). Because $\boldsymbol{y}_r = \boldsymbol{V}_r \boldsymbol{S}_r \boldsymbol{U}_r^\top \boldsymbol{x}$, $\boldsymbol{y}_r$ lies in the rank-$r$ subspace spanned by the columns of $\boldsymbol{V}_r$. Hence, letting $\boldsymbol{\mu}$ and $\boldsymbol{\Sigma}$ be, respectively, the population mean and covariance matrix for $\boldsymbol{y}$, we can exactly compute their projection onto the subspace as $\boldsymbol{\mu}_r = \boldsymbol{V}_r \boldsymbol{V}_r^\top \boldsymbol{\mu}$ and $\boldsymbol{\Sigma}_r = \boldsymbol{V}_r \boldsymbol{V}_r^\top \boldsymbol{\Sigma} \boldsymbol{V}_r \boldsymbol{V}_r^\top$ (note that diagonal components are extracted from $\boldsymbol{\Sigma}_r$ for the BN layer). For practical reasons, we compute $\boldsymbol{\mu}$ and $\boldsymbol{\Sigma}$ for each layer after training (Ioffe & Szegedy, 2015). Because $\boldsymbol{\Sigma}$ has $n(n+1)/2$ extra parameters to store, we keep $\boldsymbol{V}_{\tilde{R}}^\top \boldsymbol{\Sigma} \boldsymbol{V}_{\tilde{R}}$ instead, where $\tilde{R}$ $(1 \leq \tilde{R} < R)$ is the maximum rank to be used, reducing the number of parameters to $\tilde{R}(\tilde{R}+1)/2$. At the time of inference, we can correct the mean and variance according to the ranks in each layer. On the other hand, if a list of candidate model sizes is available in advance, we can retain the means and variances for those models as Yu & Huang (2019). We compare both methods in Section 4.

### 2.3 LEARNING

Although our scalable neural networks can operate regardless of learning methods, we propose a method to gain better performance. We simultaneously minimize losses for both the full-rank and the low-rank networks as follows.

$$\min_\Theta \frac{1}{B} \sum_{b=1}^B \left\{ (1-\lambda)\mathcal{L}(\mathcal{D}_b, \mathcal{W}, \Theta) + \lambda \mathcal{L}(\mathcal{D}_b, \widetilde{\mathcal{W}}, \Theta) \right\} \tag{3}$$

Here, $\mathcal{L}(\cdot)$ is a loss function, $\mathcal{D}_b$ is a set of training samples in a mini-batch, $B$ is the batch size, and $\lambda \in [0, 1]$ is a hyperparameter for balancing between the two losses. For this, $\mathcal{W} = \{\boldsymbol{W}^{(\ell)}\}_{\ell=1}^L$, $\widetilde{\mathcal{W}}$, and $\Theta$ are sets of $L$ weight matrices, their low-rank approximations, and other trainable parameters (e.g., biases), respectively. $\widetilde{\mathcal{W}}$ is generated from $\mathcal{W}$ via SVD and $\Theta$ is shared between the full- and low-rank networks. Therefore, the number of total paramters is not increased in learning. As shown in Figure 1(b), we additionally propagate each mini batch to a low-rank network. Because $\boldsymbol{W}_r^{(\ell)} = \boldsymbol{U}_r^{(\ell)} \boldsymbol{U}_r^{(\ell)\top} \boldsymbol{W}^{(\ell)}$, the gradient with respect to $\boldsymbol{W}^{(\ell)}$ can be computed as follows: [1]

$$\frac{1}{B} \sum_{b=1}^B \left\{ (1-\lambda)\frac{\partial \mathcal{L}(\mathcal{D}_b, \mathcal{W}, \Theta)}{\partial \boldsymbol{W}^{(\ell)}} + \lambda \boldsymbol{U}_r^{(\ell)} \boldsymbol{U}_r^{(\ell)\top} \frac{\partial \mathcal{L}(\mathcal{D}_b, \widetilde{\mathcal{W}}, \Theta)}{\partial \boldsymbol{W}_r^{(\ell)}} \right\}. \tag{4}$$

The gradients for $\Theta$ are simply computed from the $\lambda$-weighted average for those of both networks.

Since we need a single model that achieve good performance in multiple sizes which are to be selected at the inference phase, we randomly select the model size for the low-rank network at each iteration step. Specifically, a global rank ratio $Z$ is sampled from a uniform distribution $\mathcal{U}(\alpha_l, \alpha_u)$ with $0 < \alpha_l < \alpha_u \leq 1$. Then, letting $R^{(\ell)}$ be the rank of $\boldsymbol{W}^{(\ell)}$, we reduce $(1 - Z)\sum_{\ell=1}^L R^{(\ell)}$ bases across the entire network using the criterion mentioned in subsection 2.2.1. In a later section, we experimentally investigate the effects of the parameters $\lambda$, $\alpha_l$, and $\alpha_u$ in the experiment.

---

[1]In fact, $\boldsymbol{U}_r^{(\ell)}$ depends on $\boldsymbol{W}^{(\ell)}$, but we treat it as constant for simplicity.

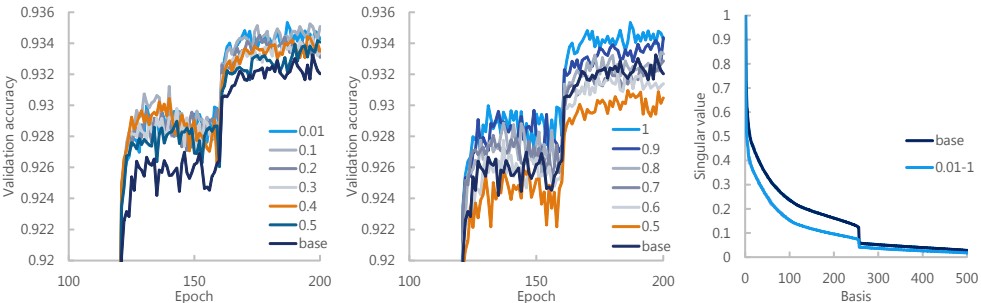

Figure 2: Training results for VGG-15 on CIFAR-10. (Left) Full-rank validation accuracy by changing $\alpha_l$ with $\alpha_u = 1.0$. (Center) Full-rank validation accuracy by changing $\alpha_u$ with $\alpha_l = 0.01$. (Right) Maximum singular value for each basis index in a full-rank model. "base" indicates normal learning as our baseline.

Arora et al. (2018); Suzuki (2019) derived the generalization error bound for DNNs under a condition that the trained network has near low-rank weight matrices. They proved that the condition contributes not only to yield a better generalization error bound for the non-compressed network but also to compress the network efficiently. That motivates our approaches: a learning scheme which aims to facilitate the performance of the low-rank networks as well as that of the full-rank network, and a compression scheme which reduces the redundant basis obtained via SVD.

## 3 RELATED WORK

**Low-rank approximation & regularization**. Compression methods based on low-rank approximation have been proposed in the literature. Denil et al. (2013); Tai et al. (2016); Ioannou et al. (2016) trained networks after factorizing the weight matrix into a low-rank form. Ioannou et al. (2016) achieved a high compression rate by factorizing a convolutional kernel of $K_w \times K_h$ into $K_w \times 1$ and $1 \times K_h$. Denton et al. (2014); Lebedev et al. (2015); Kim et al. (2016) proposed methods that use tensor factorization without rearranging the convolutional tensor into the matrix form. Yu et al. (2017) further improved the compression rate by incorporating sparseness into the low-rank constraint. Zhang et al. (2016); Li & Shi (2018) took resource-related constraints into account to automatically select an appropriate rank. Each of these methods trains a network with pre-defined ranks or compress redundant networks by applying complicated optimizations under a given target size for the model. That is, those methods would require re-training to reconfigure the models for different devices.

Kliegl et al. (2017); Xu et al. (2019) utilized trace-norm regularization as a low-rank constraint in learning the network. Wen et al. (2017) proposed a novel method called force regularization to obtain the low-rank weights. The performance of these methods depends on a hyperparamter to adjust strength of regularization. It is difficult to decide on an appropriate range for the hyperparameter in advance, meaning that selection requires trial and error to achieve a particular model size.

**Scalable neural networks**. Chen et al. (2018) represented the data flow in ResNet-type structures (He et al., 2016) as ordinary differential equations (ODEs), and proposed a Neural-ODEs, which can be used to arbitrarily control the computational cost in the depth direction. Zhang et al. (2019) also obtained scalability in the depth direction by allowing pre-defined intermediate layers to be bypassed.

Yu et al. (2019); Yu & Huang (2019) proposed slimmable networks and US-Nets, which are scalable in the width direction. Their works are closely related to ours, but there are differences in some aspects. First, since their methods directly and uniformly reduce the width for every layer, the principal components are not taken into account, and the relative importance of each layer is not considered. Second, for US-Nets in particular, they introduced a "sandwich rule" to keep the performance for an arbitrary width. However, this rule does not guarantee monotonicity of the error with increasing layer width. In the next section, we compare our method with them.

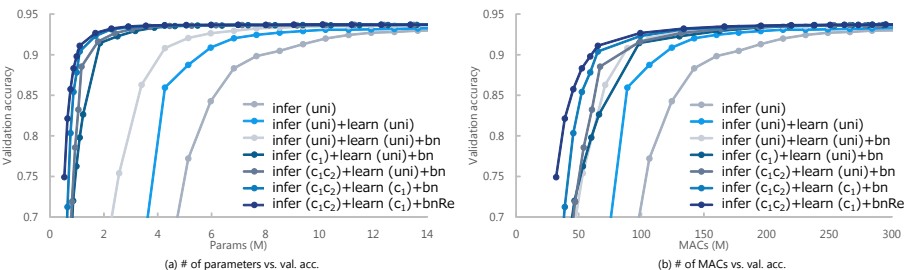

Figure 3: Inference results for VGG-15 on CIFAR-10. (a) # of parameters vs. validation accuracy. (b) # of MACs vs. validation accuracy. "+learn" indicates results with our learning method. "+bn" and "+bnRe" indicate results with our BN correction and those with recomputation, respectively. "uni", "$c_1$", "$c_2$", and "$c_1c_2$" in the bracket indicate rank selection by a uniform method, by Eq. (1), by Eq. (2), and by both, respectively. We do not apply "$c_2$" to learning because it slightly decreases the full-rank accuracy.

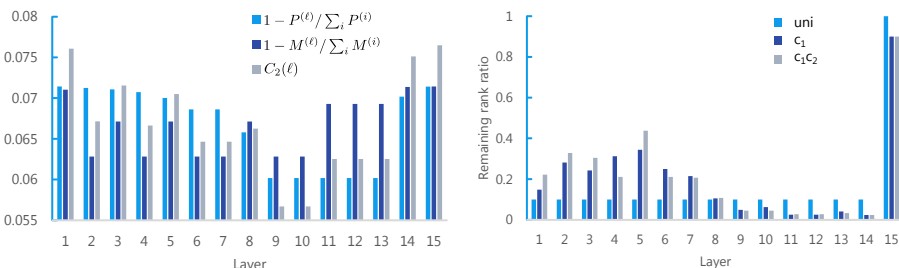

Figure 4: Results of rank selection for VGG-15 on CIFAR-10. (Left) Layer-wise importance with complexity-based criterion (normalized to sum 1). (Right) Remaining rank ratio per layer by different selection methods. "uni", "$c_1$", and "$c_1c_2$" indicate selection results by a uniform method, by Eq. (1), and by Eq. (1 & 2), respectively. We do not reduce parameters for the last fully connected layer for the uniform method because it significantly decreases accuracy.

## 4 EXPERIMENTS

We evaluate our methods on the image-classification tasks of CIFAR-10/100 (Krizhevsky, 2009) datasets using deep CNNs. The CIFAR-10/100 datasets contain $32 \times 32$ images for object recognition including 10 and 100 classes, respectively. Each dataset contains 50K images for training and 10K images for validation. We implement our method with TensorFlow (Abadi et al., 2015).

### 4.1 ABLATION STUDY

We test each component in our method on the CIFAR-10 dataset. We use the same baseline setup as in Zagoruyko & Komodakis (2016), which is summarized in Table 1 in Appendix C. Unless otherwise specified, we report the average result from 5 trials with different random seeds. We adopt a VGG-like network with 15 layers (Zagoruyko, 2015; Liu et al., 2017) [2], which we refer to as VGG-15 below.

Firstly, we evaluate our learning method for various values of the parameters $\alpha_l$ and $\alpha_u$, fixing $\lambda = 0.5$. Our method requires SVD at each iteration step, which makes it costly. To address this, we suppose that the weight subspaces are not drastically changed at each step and recompute the SVD after every two steps, reusing the results to speed up the training. We illustrate the validation accuracy of a full-rank model for different values of $\alpha_l$ (resp., $\alpha_u$) with $\alpha_u = 1.0$ (resp., $\alpha_l = 0.01$) fixed, on the left (resp., center) of Figure 2. It can be observed that smaller values of $\alpha_l$ and larger values of $\alpha_u$ are better. This can be interpreted as indicating that it is better for a full-rank model

---

[2]Since the VGG-networks are originally designed for classifying the ImageNet dataset (Deng et al., 2009), we use a smaller type than the original for the CIFAR datasets, as used by Liu et al. (2017).

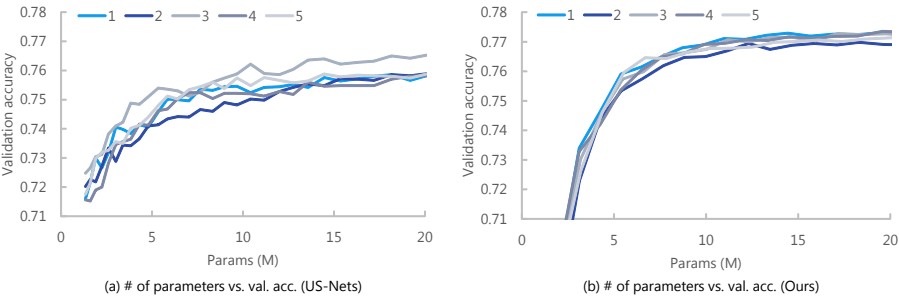

Figure 5: Scalability for ResNet-34 on CIFAR-100 dataset when using (a) US-Nets and (b) our method. We show the results for each of 5 trials with different random seeds in this figure.

to learn with various low-rank models than to learn with models biased to a specific range of ranks. Thus, we set $\alpha_l = 0.01$ and $\alpha_u = 1.0$ for the other experiments described below. On the right side of Figure 2, we show the maximum singular value for each basis index in a full-rank model [3]. We can see that our learning method obtains smaller singular values than the baseline. This implies that our learning method has an effect similar to trace-norm regularization (Kliegl et al., 2017), suggesting that we can suppress the errors produced by reducing the bases.

Next, we evaluated the performance of our inference method for various model sizes. In Figure 3, we illustrate the inference results for validation data with various number of parameters and MACs. In the figure, "infer (uni)" indicate the results obtained by uniformly reducing the basis in each layer. Concretely, with a global rank ratio $G \in (0, 1]$, we reduce $(1 - G)R^{(\ell)}$ bases in order from the one corresponding to the smallest singular value. Despite the method being simple, the accuracy changes almost smoothly, and it can be confirmed that there the accuracy scales against changes in the model size. This can be considered as due to the monotonicity of errors, which is formalized in Proposition 2.1. Additionally, the performance is also improved with our learning method by applying uniform rank selection and by using our BN correction. Furthermore, the performance with respect to the parameters is improved when we apply the error- and complexity-based criteria for rank selection to both inference and learning (in the figure, "$c_1$" and "$c_1 c_2$"). However, the performance with respect to the MACs is dropped by changing the rank selection from uniform ("uni") to error-based ("$c_1$"). As shown on the left side of Figure 4, it is more effective for decreasing MACs to reduce the parameters in shallower layers, which have large feature maps. However, the error-based criterion tends to reduce the parameters in deeper layers because those tend to be low rank. When both criteria are applied (in the figure, "$c_1 c_2$"), the performance is also improved for the MACs. We show the rank-selection results for different criteria on the right side of Figure 4. It can be confirmed that the ranks are decreased for 4, 6, 7, 9, and 10 layers which have large MACs in the case with both criteria ("$c_1 c_2$") relative to the case with only the error-based criterion ("$c_1$"). For the BN correction, our method is effective, but it is better with a method that recomputes the means and variances for given ranks ("bnRe"). Because our method is layer-by-layer correction, this probably occurs because our method cannot fully correct for the inter-layer gap, with the statistics of the deep layer changing due to the reduction of rank in the shallow layer.

Additionally, we investigate the effect of a parameter $\lambda$. We evaluate the validation accuracy with respect to the number of paramters for $\lambda \in \{0.1, 0.2, 0.3, 0.4, 0.5\}$ with VGG-15 and ResNet-34 on the CIFAR-10/100 datasets. The results are shown in Figure 7 in Appendix D. We consider that there is a trade-off between the performance of full- and low-rank models, which depends on $\lambda$.

## 4.2 COMPARISON WITH SLIMMABLE NETWORKS

We compare our method with slimmable networks (Yu et al., 2019) and US-Nets (Yu & Huang, 2019) in terms of performance on the CIFAR-10/100 datasets. We adopt VGG-15 and ResNet-34 (He et al., 2016). We implement the models based on the Yu's code, available at `https://github.com/JiahuiYu/slimmable_networks` (written in PyTorch (Paszke et al., 2017)).

---

[3]We let $\sigma_{ij}$ be a singular value for a basis $j$ in a layer $i$ and then compute $\max_i (\sigma_{ij})$. For layers with lower ranks, we simply fill the missing part with zeros.

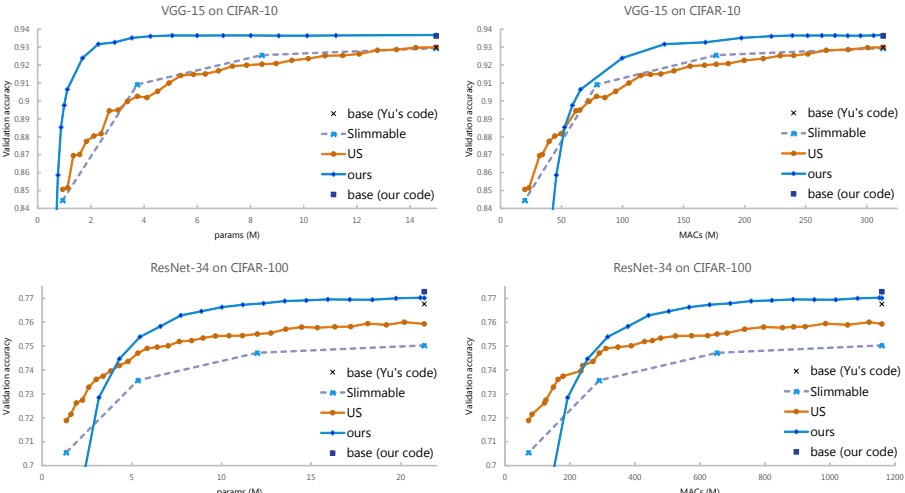

Figure 6: Comparisons with slimmable networks (Yu et al., 2019) and US-Nets (Yu & Huang, 2019) for VGG-15 and ResNet-34 on CIFAR-10/100 datasets. (Left) # of parameters vs. validation accuracy. (Right) # of MACs vs. validation accuracy.

US-Nets is trained with 2 random widths between the lower and upper width and in-place distillation (Yu & Huang, 2019), then BN-calibration (Yu & Huang, 2019) is applied to each of the slimmable networks and US-Nets after training. For our method, we incorporate all components into the comparisons and adopt BN correction with recomputation. We train the models using $\lambda = 0.4$ and the same setup as in the previous subsection. In the following, we report the results for models after the last iteration in training.

First, we compare the scalability of ResNet-34 on the CIFAR-100 dataset. We illustrate the inference results over various numbers of parameters for 5 models trained with different random seeds in Figure 5. The results in the figure show that US-Nets are unstable, which is a problem for practical use. This instability is because US-Nets do not have monotonic error changes in each layer, a property that our method ensures. Next, we show the results for comparison of VGG-15 on CIFAR-10 and ResNet-34 on CIFAR-100 in Figure 6. The notations "base (Yu's code)" and "base (our code)" indicate the baseline results obtained by the Yu's code and our code with the same setup. Our baseline is slightly better than the Yu's baseline. We consider this to be due to differences in the framework. Comparing the results with those for VGG-15 on CIFAR-10, our method tends to be more accurate in terms of the number of parameters than in terms of the number of MACs. Since deep layers have more parameters than shallow layers, the rank in deep layers tends to be lower than in shallow layers, resulting in more paramters reduced in deep layers by our method. In contrast, US-Nets reduce the width uniformly across layers, which may contribute to reducing the number of MACs. However, reducing the number of MACs does not necessarily lead to cut the inference cost dominantly, depending on the target device (Yang et al., 2018). Although we only consider the number of parameters and MACs as the complexity metrics in this paper, other metrics such as memory footprint, memory access cost, and runtime latency should be taken into account for validating the effectiveness in practical use (Tan et al., 2019; Sandler et al., 2018; Dai et al., 2019).

We can see that the accuracy of our method is lower than that of US-Nets when the compression rate is extremely high. Our method uses SVD and reduces the bases, which means it does not change the number of inputs and outputs (i.e., the in and out dimensionalities). Because the number of parameters in each layer is $(m + n)r$, it decreases linearly with respect to the rank. US-Nets reduce both input and output dimensionality, meaning that the number of parameters is decreased at a quadratic rate. This makes it easier for US-Nets to achieve extremely high compression. However, our method is better in larger regimes. In particular, for ResNet-34 on CIFAR-100, the performance of slimmable networks and US-Nets on the full-size model are decreased, while our method does not decrease performance much. We illustrate additional comparison results in Figure 8 in Appendix D and give an analysis of per-layer error in Appendix E.

## 5 CONCLUSIONS

We proposed a novel method that enables DNNs to flexibly change their size after training. We used to factorize the weight matrix for each layer into two low-rank matrices after training the DNNs. By changing the rank in each layer, our method can scale the model to an arbitrary size. We introduced simple criteria for characterizing the importance of each basis and layer; these are the error- and complexity-based criteria. Those criteria enabled effectively compressing models without introducing much error. In experiments on multiple image-classification tasks using deep CNNs, our method exhibited good performance relative to that of other methods.

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

APPENDICES

## A  PROOF OF PROPOSITION 2.1

As $\boldsymbol{y} = \boldsymbol{y}_R$ by definition, $\|\boldsymbol{y} - \boldsymbol{y}_R\|^2 = 0$. To prove $\|\boldsymbol{y} - \boldsymbol{y}_1\|^2 \geq \cdots \geq \|\boldsymbol{y} - \boldsymbol{y}_r\|^2 \geq \|\boldsymbol{y} - \boldsymbol{y}_{r+1}\|^2 \geq \cdots \geq \|\boldsymbol{y} - \boldsymbol{y}_R\|^2$, we show that $\|\boldsymbol{y} - \boldsymbol{y}_r\|^2 - \|\boldsymbol{y} - \boldsymbol{y}_{r+1}\|^2 \geq 0$ for $1 \leq r \leq R - 1$.

*Proof.*

$$
\begin{aligned}
\|\boldsymbol{y} - \boldsymbol{y}_r\|^2 - \|\boldsymbol{y} - \boldsymbol{y}_{r+1}\|^2 &= \| \left( \boldsymbol{I}_n - \boldsymbol{V}_r \boldsymbol{V}_r^\top \right) \boldsymbol{y} \|^2 - \| \left( \boldsymbol{I}_n - \boldsymbol{V}_{r+1} \boldsymbol{V}_{r+1}^\top \right) \boldsymbol{y} \|^2 \\
&= \boldsymbol{y}^\top \left( \boldsymbol{I}_n - \boldsymbol{V}_r \boldsymbol{V}_r^\top \right) \boldsymbol{y} - \boldsymbol{y}^\top \left( \boldsymbol{I}_n - \boldsymbol{V}_{r+1} \boldsymbol{V}_{r+1}^\top \right) \boldsymbol{y} \\
&= \boldsymbol{y}^\top \left( \boldsymbol{V}_{r+1} \boldsymbol{V}_{r+1}^\top - \boldsymbol{V}_r \boldsymbol{V}_r^\top \right) \boldsymbol{y} \\
&= \boldsymbol{y}^\top \left( \boldsymbol{v}_{r+1} \boldsymbol{v}_{r+1}^\top \right) \boldsymbol{y} \\
&= \left( \boldsymbol{v}_{r+1}^\top \boldsymbol{y} \right)^2 \geq 0
\end{aligned}
\tag{5}
$$

$\square$

Here, $\boldsymbol{I}_n$ indicates the identity matrix with size $n \times n$.

## B  AN ALGORITHM FOR RANK SELECTION

---
**Algorithm 1** Rank selection
---
**Input:** A network with weight matrices $\mathcal{W} = \{\boldsymbol{W}^{(\ell)}\}_{\ell=1}^L$ in which $\boldsymbol{W}^{(\ell)}$ has rank $R^{(\ell)}$.
**Input:** A criterion $C$ and the target model size $T$ (e.g., # of parameters and MACs).
**Output:** A set of tuples $\mathbb{S}$ which contains indices of layer and basis.
 1: $\mathbb{S} \leftarrow \emptyset$
 2: **for** $\ell = 1, \ldots, L$ **do**
 3:     **for** $k = 1, \ldots, R^{(\ell)}$ **do**
 4:         Compute $C(\ell, k)$.
 5:         $\mathbb{S} \leftarrow \mathbb{S} \cup \{(\ell, k)\}$.
 6:     **end for**
 7: **end for**
 8: Arrange elements in $\mathbb{S}$ in ascending order of $C(\ell, k)$.
 9: Delete the first $d$ elements in $\mathbb{S}$ to satisfy the model size $T$.
10: **return** $\mathbb{S}$
---

## C  BASELINE SETUP OF EXPERIMENTS ON CIFAR-10/100 DATASETS

Table 1: Baseline setup of experiments on CIFAR-10/100 datasets.

| | |
|---|---|
| Preprocess | Per-channel standardization (mean, std.)
  CIFAR-10 : (0.4914, 0.4822, 0.4465), (0.2470, 0.2435, 0.2616)
  CIFAR-100: (0.5071, 0.4865, 0.4409), (0.2673, 0.2564, 0.2762) |
| Data augmentation | Random cropping $32 \times 32$ after zero-padding 4 pixels
Random horizontal flipping ($p = 0.5$) |
| Batch size / Epochs | 128 / 200 |
| Optimizer | SGD with Nesterov momentum ($\mu = 0.9$) |
| Learning rate | Initialized to 0.1, multiplied by 0.2 at 60, 120, and 160 epochs |
| $L_2$ regularization | 0.0005 |
| Initializer | He-Normal (He et al., 2015) for weights, 0 for biases |
| BN | $\epsilon = 1.0 \times 10^{-5}, momentum = 0.9$. Initialize $\gamma = 1$ and $\beta = 0$ |
| GPUs | 1 |

# D ADDITIONAL RESULTS FOR VGG-15 AND RESNET-34 ON CIFAR DATASETS

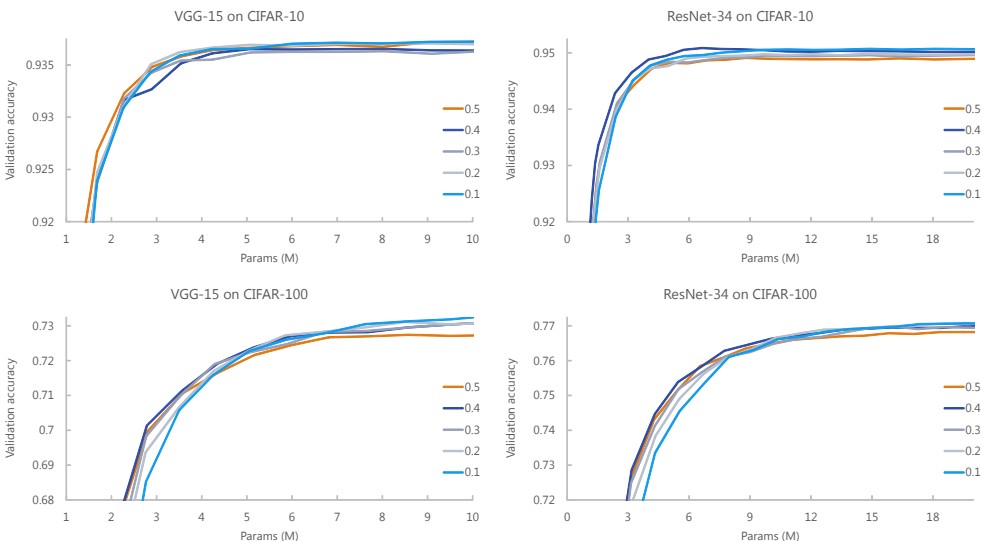

Figure 7: The effect of a hyper-parameter for balancing the losses ($\lambda$). Validation accuracies are evaluated with $\lambda \in \{0.1, 0.2, 0.3, 0.4, 0.5\}$ for VGG-15 and ResNet-34 on CIFAR-10 / 100 datasets.

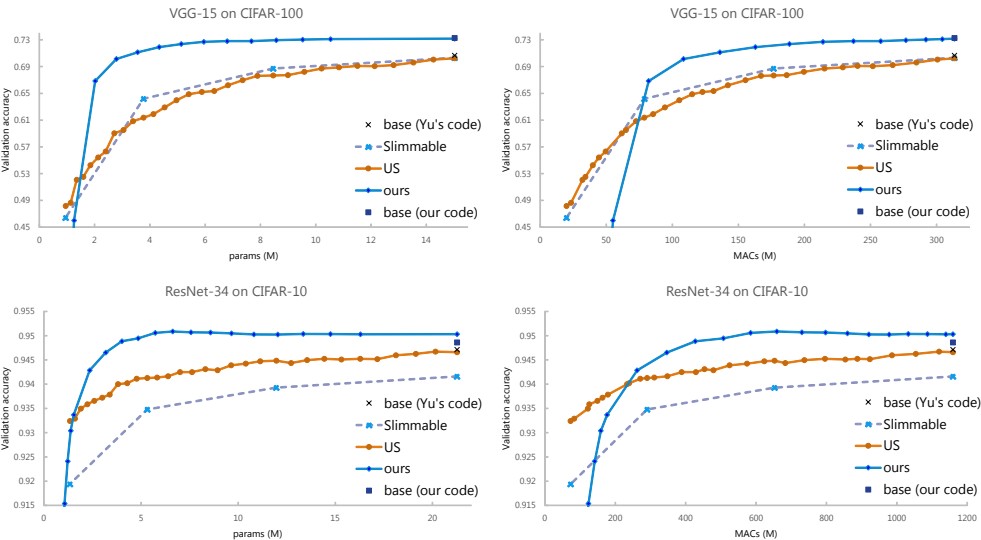

Figure 8: Comparisons with slimmable networks (Yu et al., 2019) and US-Nets (Yu & Huang, 2019) for VGG-15 and ResNet-34 on CIFAR-10/100 datasets. (Left) # of parameters vs. validation accuracy. (Right) # of MACs vs. validation accuracy.

# E AN ANALYSIS OF PER-LAYER ERROR

We train VGG-15 on the CIFAR-10 dataset with US-Nets (Yu & Huang, 2019) and our method. First, we illustrate learned weights in Figure 9. For comparison, we show a weight matrix $\boldsymbol{W}$ for US-Nets and its factorized form $\boldsymbol{US}$ obtained via SVD for our method. The rank of $\boldsymbol{W}$ in the 1st convolutional layer is 27, and thus only 27 convolutional tensors are shown for our method in the upper part of the figure. Since $\boldsymbol{W}$ becomes low-rank after training, there is only a little number of

dominant bases which correspond to higher singular values in $\boldsymbol{S}$. For US-Nets in particular, we can see that weight coefficients of large absolute values are concentrated in lower channels. It can be considered that US-Nets implicitly attract important features into lower channels because US-Nets reduce channels in order from the one with higher indices. Therefore, it can be expected the errors induced by reducing channels are partially suppressed for US-Nets.

Next, we compute the sum of squared error: $\sum_{n=1}^{N} \|\boldsymbol{Y}_n^{(\ell)} - \widetilde{\boldsymbol{Y}}_n^{(\ell)}\|_F^2 / \sum_{n=1}^{N} \|\boldsymbol{Y}_n^{(\ell)}\|_F^2$ over the validation dataset, where $N$, $\boldsymbol{Y}_n^{(\ell)}$, and $\widetilde{\boldsymbol{Y}}_n^{(\ell)}$ are the number of validation samples, an output feature map of layer $\ell$ for the full-size network, and that for the compressed network, respectively. With compressing the entire network from $100\%$ to about $5\%$ in terms of the total number of parameters, we compute the squared error with respect to the number of parameters in each layer as shown in Figure 10. For our method, we reduce the bases of weight marices (i.e., columns in $\boldsymbol{U}^{(\ell)}\boldsymbol{S}^{(\ell)}$) according to the criterion $C_1(\ell, k)$ $C_2(\ell)$ and compute $\widetilde{\boldsymbol{Y}}^{(\ell)}$ with a low-rank weight matrix $\boldsymbol{W}_r^{(\ell)}$ in each layer. For US-Nets, since the dimensionality of $\widetilde{\boldsymbol{Y}}^{(\ell)}$ is decreased with reducing channels for every layer, we fill the missing part with zeros. It can be confirmed the squared errors by our method are suppressed more than that by US-Nets in Figure 10. We consider this is because our method do not directly reduce the channel as in US-Nets but instead reduce the redundant basis in the column space of the weight matrix, which prevents the feature map in each layer from losing important features (i.e., principal components).

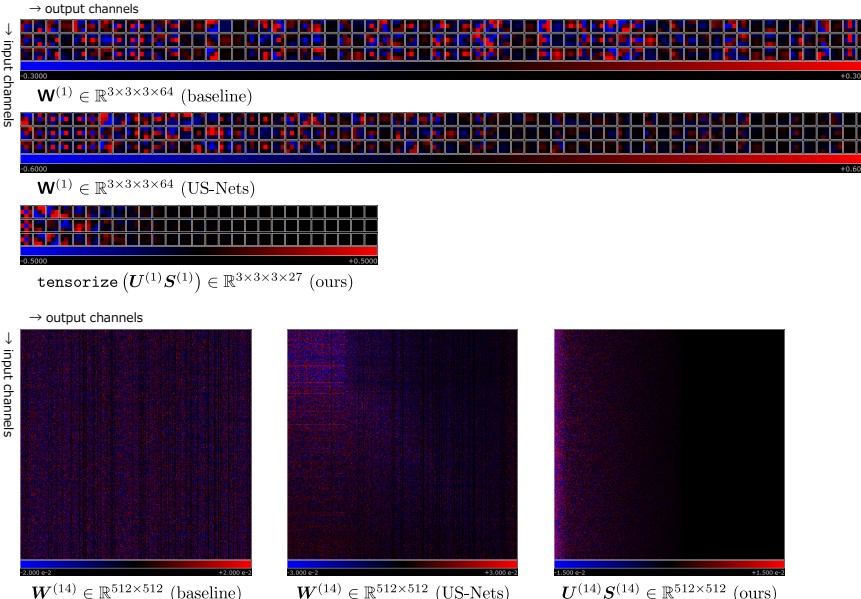

Figure 9: Visualizations of weight coefficients of VGG-15 trained on the CIFAR-10 dataset. (Upper) Weight tensors in the 1st convolutional layer. (Lower) Weight matrices in the 14th fully connected layer. "baseline" indicates normal learning.

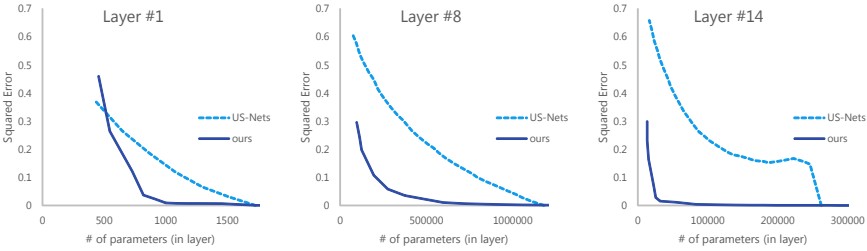

Figure 10: # of parameters vs. the sum of squared error for the 1st, 8th, and 14th layers of VGG-15 trained on the CIFAR-10 dataset.

