# OpenReview forum: "Scalable Deep Neural Networks via Low-Rank Matrix Factorization"
_ICLR.cc/2020/Conference — Reject_

### Official Review · AnonReviewer3 · 2019-10-23
**Official Blind Review #3**

**Rating:** 3

**Review:**

This paper propose to compress deep neural network with SVD decomposition, which however has been published 6 years ago in this paper to decompose FC layers:
Xue, Jian, Jinyu Li, and Yifan Gong. "Restructuring of deep neural network acoustic models with singular value decomposition." Interspeech. 2013.
For convolutional layers, Tucker decomposition, which is high-order SVD, is apparently a better choice. (Kim et al. (2016)) They should at least compare their method with this one.

In their deduction of full-rank-low-rank model joint training,  W_r^{(l)} = U_r^{(l)}{U_r^{(l)}}^T W^{(l)} is used without any explanation.  Since U_r^{(l)}{U_r^{(l)}}^T = I_r, W_r^{(l)} will be first r rows of W^{(l)} or first r cols of W^{(l)}.  It can not be treated as approximation of W^{(l)}.  In other words, the foundation of their training is wrong, which makes their experimental results unconvincing.

They conduct their experiments with CIFAR-10/100, which is too small for VGG-15 and ResNet-34.  A larger dataset would be better.

The writing of this paper is not good. For example:
1. In proposition 2.1, what does "y" represent.
2. For Error-based criterion, what is its difference with selecting rank according to singular values?
3. What is P^{(i)} and M^{(i)} in complexity-based criterion?

In conclusion, i will give a weak reject.

**Experience Assessment:**

I have published one or two papers in this area.

**Review Assessment: Checking Correctness Of Derivations And Theory:**

I assessed the sensibility of the derivations and theory.

**Review Assessment: Checking Correctness Of Experiments:**

I assessed the sensibility of the experiments.

**Review Assessment: Thoroughness In Paper Reading:**

I read the paper thoroughly.

---

> ### Author Response · Authors · 2019-11-15
> **Reply to reviewer #3**
>
> Thank you for your thoughtful comments.
>
> We will investigate or compare with those methods you suggested and also consider to do experiments on larger datasets as our future works.
> We have revised some notations according to your suggestions.
>
> >In their deduction of full-rank-low-rank model joint training, .....
> $U_r^{(\ell)}{U_r^{(\ell)}}^T$ is not $I_r$ but a projection matrix onto the rank-$r$ subspace (may be confusing with ${U_r^{(\ell)}}^T U_r^{(\ell)}=I_r$).

---

### Official Review · AnonReviewer2 · 2019-10-23
**Official Blind Review #2**

**Rating:** 1

**Review:**

This paper proposes to reshape the weights of the layers of deep neural networks and parametrize them with a low-rank matrix decomposition (SVD). The rank is optimized using two criterion (error-based and complexity-based). Since the decomposition is applied post-hoc, the authors propose to correct the parameters of the batch norm analytically. The authors propose to jointly optimize a loss on the full network and the low-rank version. Experiments are done on CIFAR 10 an 100 with a VGG-15 and ResNet-34 architecture.

Some of the ideas are interesting and would be worth developping further. However, the paper in the current state cannot be accepted for the following reasons: (1) the novelty is low, this very type of decomposition is already widely studied (2) the paper is not clear as to what the contributions are, and why they are justified, theoretically or empirically, (3) the review and comparison with the state-of-the-art is lacking and (4) the experimental setup is simplistic and not convicing. (5) overall the paper is imprecise.


Main comments

* Applying SVD to the matricized weights of deep neural networks is not new. Actual contributions need to be separated from existing works.
* The related word needs to be reviewed. Many references are missing. In particular, the proposed method could be considered as a special case of tensor based methods.
* The references that *are* listed in the related work are not properly reviewed: the authors aim to not compare with them claiming that they require re-training. Lebedev et al provide a method that works both for end-to-end training or post-hoc, by applying tensor decomposition to the trained weights. Fine-tuning is optional and done to recover performance.
* How was the complexity-based criterion obtained? Why use both M and P, since M includes P? How do the proposed criterion compare to simple measures, e.g. explained variance?
* The authors should compare with other compression techniques: layer-wise compression (e.g. Lebedev et al, Speeding-up convolutional neural networks using fine-tuned cp-decomposition, ICLR 2015) or full network compression (e.g. Kossaifi et al, T-Net: Parametrizing Fully Convolutional Nets with a Single High-Order Tensor, CVPR 2019).

Missing experiments

* The proposed learning loss needs to be compared with the original one to demonstrate any potential performance improvement. Currently, it is not clear whether it is actually helping. In other words, there should also be a comparison with \lambda = 1 or 0.
* Does the proposed loss have an effect on the selected rank? On the actual rank of the weights? On the distribution of the eigenvalues?
* The BN correction needs to be experimentally motivated: since the network is trained with a loss that incorporates the low-rank network, is that needed? Does the propose loss affect performance? How does performance change with and without that BN correction?
* Experiments on CIFAR 10-100 is not sufficient to be convincing, the authors should ideally try a more realistic, large scale dataset, e.g. ImageNet.
* VGG-15 is not convincing to show-case compression, as more than 80% of the parameters are in the fully-connected layer
* The authors assume that the SVD decomposition of the weights does no change significantly at each step: is there any empirical evidence supporting this assumption? This most likely depends on the experimental setup (batch-size, learning rate, etc).


**Experience Assessment:**

I have published in this field for several years.

**Review Assessment: Checking Correctness Of Derivations And Theory:**

I carefully checked the derivations and theory.

**Review Assessment: Checking Correctness Of Experiments:**

I carefully checked the experiments.

**Review Assessment: Thoroughness In Paper Reading:**

I read the paper at least twice and used my best judgement in assessing the paper.

---

> ### Author Response · Authors · 2019-11-15
> **Reply to reviewer #2**
>
> Thank you for your thoughtful comments. We reply in order.
>
> >1, 2, 3, and 5. Thank you for suggesting. We will investigate or compare with those methods as our future works.
>
> >4. It is based on heuristics that the proportion of # of parameters and MACs in each layer could be a selection criterion for efficiently reducing the complexity of entire network. It was experimentally better to use both $M$ and $P$ than to use either.
>
> >6. A corresponding result for $\lambda = 0$ (regular loss) is shown in Figure 3 ("infer (uni)"), which is good for full-rank model (right-most point in Figure 3) but it is poor for reduced models. In the case of $\lambda = 1$, for which we minimize a loss only for low-rank network whose ranks are randomly determined in each iteration, but the result was poor. We consider this is because randomness is too strong to learn a model.
>
> >7. At least, the method has an effect on the distribution of the singular values. As shown in Figure 2 (right most), singular values with the proposed loss are smaller than that with regular loss, meaning that approximation errors could be suppressed more than regular loss.
>
> >8. BN correction is needed only for inference. As shown in Figure 3, the proposed BN correction is not better in terms of accuracy than simply computing mean & var for each model after training. However, for recomputation, the model sizes must be fixed in advance and the computation is required for every model to be used. Our method requires the computation only one time for full-rank model and can analytically produce mean & var for the model with any rank.
>
> >9. Thank you for suggesting. We will take it into consideration.
>
> >10. Different from an original VGG-16, VGG-15 has only one FC layer (other than the last one for classification), which has only 512 nodes. Therefore, the proportion of # of parameters in the FC layer is low.
>
> >11. In the preliminary experiment on the CIFAR datasets, we confirmed that the performance gap is negligible for the interval of 2.

---

### Official Review · AnonReviewer1 · 2019-10-24
**Official Blind Review #1**

**Rating:** 1

**Review:**

This paper proposes a method to modify the computing requirements of a trained model without compromising the accuracy and without the need for retraining (with new requirements). To this end, the algorithm focuses on factorizing the models and uses a 2-branch training process to train low-rank versions of the original model (to minimize the accuracy drop).

Different from other approaches, the algorithm claims to exploit the importance of each layer when reducing the compute.



Comments:
- Figure 2 and related numbers are slightly misleading. The paper focuses on CNN while these numbers and the figure is for FC. M changes significantly when using convolutions. It would be great to clarify this all over the text as M increases signigicantly when using (at least) 3x3 convolutions.

- One missing thing for me is taking into account the latency rather than the number of parameters. While factorization may reduce the number of parameters (considering the rank is sufficiently low), the number of layers and therefore data movements increases and so does the latency. Some analysis on this can be found in [1] where the paper trains a network with low-rank promoting regularization. I missed having [1] and other similar approaches in the related work and how the proposed method compares to those directly promoting low-rank solutions.

- The approach would be sounded if the algorithm does not need to work on the factorized version of the layer. That would bring direct benefits to inference time.

- On the algorithmic side, it is not clear to me how the "2-branches" are trained and what parameters are shared. This seems to involve more compute, right? How is this better than aiming at the lowest-rank possible?

- The complexity-based criterion is interesting although only uses FLOPS as a proxy. How would this translate in practice when the latency is not directly represented by the FLOPS (given the parallelization).


- During the learning, it is not clear to me how the process is implemented and the scalability of this approach. The paper suggests computing SVD per iteration is infeasible. How many iterations are between SVD? and how results are reused. How this is different from [1] where the authors used truncated SVD to promote low-rank every epoch?

- I need clarification on the need of training full-rank and low-rank (end of page 4). If full-rank does not actually provide better accuracy (see [1]), then, why do we need to rely on that?

- Section 3 focuses on works relying on retraining. It would be nice to see how the proposed method compares to those not considering retraining to improve accuracy.

- Not clear to me what is the take-home message with Figure 4. Is the resulting rank per layer enough to represent a big compression? Why not compressing the last layer. the compression is limited as the factorization does not promote sparsity when combining the basis (Sparsity on V). Thus, the size after factorization is the same as the original.

- The BN correction does not seem to contribute. Experiments suggest: our method is effective. Not very appealing as a contribution.


Minor details:
- x is used as a single input (page 2) and as an entire dataset (page 3)?

- How do we set the \alpha values?





[1] Compression-aware training of DNN, Alvarez and Salzmann. NeurIPS 2017

**Experience Assessment:**

I have published in this field for several years.

**Review Assessment: Checking Correctness Of Derivations And Theory:**

I assessed the sensibility of the derivations and theory.

**Review Assessment: Checking Correctness Of Experiments:**

I assessed the sensibility of the experiments.

**Review Assessment: Thoroughness In Paper Reading:**

I read the paper thoroughly.

---

> ### Author Response · Authors · 2019-11-15
> **Reply to reviewer #1**
>
> Thank you for your thoughtful comments. We reply in order.
>
> >1. We have described in the revised version that $m$ and $n$ are $K_w K_h C_{in}$ and $C_{out}$, respectively, for CNNs.
>
> >2. As you commented, latency would be better performance measure for practical evaluation, which remains as one of future tasks.
>
> >4. Because we input each mini-batch to full- and low-rank networks, the computation time for forward and back prop. will be increased. A weight matrix of each layer in the low-rank network is generated by applying SVD to the full-rank network and other parameters are shared with the full-rank network. Thus, the number of total parameters is not increased. We revised Figure.1 to better explain our method.
>
> >5. Currently, we don't have appropriate answer but it may depend on the target device.
>
> >6, 7. Two iterations between SVD. This means that $U$, $S$, and $V$ in low-rank network are updated once in every two iterations while $W$ in full-rank network are updated every iteration. A method in [1] uses trace-norm regularizer to obtain low-rank weight matrices. We consider it is only suitable for a resulting low-rank model. Therefore, the performance of full-rank model is not explicitly compensated while our method explicitly does by minimizing losses for both of full- and low-rank network.
>
> >8. Thank you for suggesting. We will investigate or compare with those methods as our future works.
>
> >9. We did not compress the last FC layer with uniform reduction ("uni") but we did with our criterion ("c1" and c1c2""). Please see the right side of Figure 4 (it is slightly different). We consider this is because the last FC layer is important for classification.
>
> >10. As shown in Figure 3, the proposed BN correction is not better in terms of accuracy than simply computing mean & var for each model after training. However, for recomputation, the model sizes must be fixed in advance and the computation is required for every model to be used. Our method requires the computation only one time for full-rank model and can analytically produce mean & var for the model with any rank.
>
> >11. We use $x$ as a single input in each layer. We revised the notation in page 3.
>
> >12. It should be experimentally determined as shown in Figure 2.

---

### Public Comment · ~Yuhui_Xu2 · 2019-10-10
**About the training procedure**

Hi,

Thanks for sharing your interesting work!
The biggest contribution of this work is the new scalable low rank decomposition. While I think you may miss a relevant reference Trained Rank Pruning[1] which proposes a similar procedure(embeds the low-rank decomposition in the training).

[1]Trained Rank Pruning for Efficient Deep Neural Networks https://arxiv.org/abs/1812.02402v2

---

> ### Author Response · Authors · 2019-10-21
> **Thank you for your comments.**
>
> The paper [1] proposes a novel training method to improve the performance of low-rank network:
> - apply SVD in training for every $m$ iterations and reduce ranks according to a fixed ratio $e \in [0, 1]$.
> - impose trace norm regularization with a parameter $\lambda$ to facilitate low-rankness.
>
> The paper [1] shows theoretical guarantee of convergence to a specific rank, where the resulting rank depends on $e$. That is, the rank of network is fixed after training and thereby is not assumed to be flexibly changed.
>
> Our training method explicitly minimizes losses for both of full- and low-rank network, which is designed not only to keep the performance of full-rank network but also to improve that of multiple low-rank networks (whose ranks are randomly determined in training). We consider the method helps network to perform well for multiple ranks to be used after training.
>
> Anyway, we will cite it as related work.
> We thank you again.

---

### Author Response · Authors · 2019-11-15
**Reply to all reviewers**

We would like to thank all the reviewers for their careful comments.
First of all, let us comment comprehensively.
We will reply to individual comment not covered by this post.

# Contributions
According to the comments from the reviewers, the novelty of our method has questioned due to utilization of classical low-rank matrix factorization techniques (i.e. SVD).
However, our method is not for compaction as in the literature [1], which aim to compress the model to a specific size, but for scalable usage in which the size of DNNs are changed without retraining.
Although the researches on this purpose (scalability of the model) have been done by Yu et al. (2019), there are some points to be improved.
We propose a different approach based on low-rank matrix factorization, to the best of our knowledge, which is novel at least for this purpose (scalability of the model).
On the algorithmic side, we believe that there is a novelty in our training procedure that explicitly minimizes losses for both of full- and low-rank network.
The main contributions of our work are as follow:
1. In contrast to a work by Yu et al. (2019), we do not directly reduce the width but instead reduce the redundant basis obtained via SVD, which prevents the feature map in each layer from losing important features.
2. We propose a training method, which is designed not only to keep the performance of full-rank network but also to improve that of multiple low-rank networks (to be used at the inference phase).

# Relation to low-rank based compression methods
While low-rank compression methods achieve good performance with a model of specific size, we need a single model that achieve good performance in multiple sizes which are to be selected at the inference phase.
We consider that our training method (in the second contribution) is effective to achieve the purpose and is different from other methods that impose a certain low-rankness in training [1].
In addition, although we used simple channel decomposition by SVD, the proposed scheme does not depend on the decomposition method.
Therefore, the other decomposition methods such as spatial decomposition (Ioannou et al., 2016) and tensor decomposition (Kim et al., 2016) can be applied.
We will investigate or compare with those methods as our future works.

We have revised the paper to better explain the details of our concept (Figure 1 in particular) and added ablation studies for a contribution 1 in appendix E.

[1] Compression-aware training of DNN, Alvarez and Salzmann. NeurIPS 2017.

Thank you.
Authors.

---

### Decision · Program_Chairs · 2019-12-19

**Decision:**

Reject

**Comment:**

The proposed paper presents low-rank compression method for DNNs. This topic has been around for a while, so the contribution is limited. Lebedev et. al paper in ICLR 2015 used CP-factorization to compress neural networks for Imagenet classification; in 2019, the idea has to be really novel in order to be presented on CIFAR datasets. The latency is not analyzed.
So, I agree with reviewers.